# A Qualitative Study on How Younger Women Experience Living with an Ostomy

**DOI:** 10.3390/ijerph20095627

**Published:** 2023-04-24

**Authors:** Andrea Emilie Mørkhagen, Line Nortvedt

**Affiliations:** Department of Nursing and Health Promotion, Faculty of Health Sciences, OsloMet, Oslo Metropolitan University, 0130 Oslo, Norway

**Keywords:** stoma, ostomy, younger women, experiences, information

## Abstract

There is a growing demand that ostomy patients receive more systematic and individualised follow-up by ostomy nurses. The purpose of the study was to explore how younger women experience everyday life after an ostomy and to map what healthcare personnel can do to ensure that the patient group can feel safe and looked after. This qualitative study included four younger women who had a stoma fitted. Individual in-depth interviews were conducted, and two participants were interviewed twice. The findings resulted in three main themes: (1) The importance of follow-up and information from healthcare personnel, (2) Experience with illness and freedom in everyday life and (3) Self-image and social relationships. We found that time to prepare before surgery and learning to live with the stoma provide a good basis for handling the new everyday life with a stoma. We conclude that ostomy nurses provide support and security to those undergoing ostomy operations. Healthcare professionals should focus on providing individually tailored information to ensure that patients are receptive to the information being shared with them. Having parts of a bowel removed can be experienced as relief, especially when the disease has previously contributed to poor self-image and social isolation.

## 1. Introduction

According to the Norwegian Association of Ostomy, Reservoir and Intestinal Cancer (NORILCO), in 2020, 22,853 people availed of ostomy equipment on a blue prescription, which gives an indication of the occurrence of ostomy in Norway [1]. Some people get a stoma because of bowel or urinary tract cancer, trauma, or inflammatory diseases, such as ulcerative colitis or Crohn’s disease. Some people experience getting a temporary stoma only for a certain period, often as relief, whereas others get a stoma as a permanent solution [1].

Ostomy patients face several challenges for example, stoma leakage can cause challenges that affect leisure time in social contexts and at work [2,3]. Ostomy patients have no control over when faeces or air comes out of the stoma. For some patients, this means that their self-image is affected in a negative direction, as there is an expectation from the environment in society that one should always be clean and that noises and smells coming out of one’s body should not be noticed by others [4]. A new, unfamiliar everyday life with a stoma that involves changes in body and self-image can cause major upheavals that take time to cope with and adjust to [5].

According to NORILCO [1], the starting point for the installation of a stoma varies greatly for each patient. For those who receive a stoma without warning, for example, cancer patients, it can often come as a shock because the patient goes from living a normal everyday life as a healthy person to receiving a serious diagnosis followed by stoma surgery within a few days. For others, it can be a positive experience; for example, in patients with chronic inflammatory disease, where their everyday life consists of intense pain and frequent visits to the toilet with diarrhoea, having a stoma can make everyday life more predictable and less painful [6]. The challenges of living with a stoma can include, for example, concerns about self-image and sexuality, as well as questions about pregnancy, diet, etc. [4].

There is a growing demand that ostomy patients receive more systematic and individualised follow-up by ostomy nurses [7]. A narrative literature review showed that quality of life among ostomy patients was influenced by, among other matters, having access to an ostomy nurse, self-efficacy, skin complications, poor body image, worry related to intimate and sexual problems [8].

A Chinese descriptive cross-sectional study showed that the quality of life of patients with a stoma was generally of a medium level, and the degree of psychosocial adaptation was of particular importance for the quality of life in this patient group [9]. The study conclude that nurses’ attention aimed at increasing the amount of knowledge and skills related to the stoma, as well as help to accept the stoma, would enhance their quality of life [9].

Open communication can help patients talk about their negative feelings related to the ostomy, and health professionals should encourage patients to express their feelings and seek advice based on their needs [10]. According to Ayaz-Alkaya (2019), ostomy patients are concerned about facing challenges in social contexts and experience negative reactions regarding body and self-image [11]. Moreover, fatigue, psychosocial problems, pain, challenges in carrying out everyday activities, fear of leakage and sexual problems were experienced by patients living with ileostomies in Denmark [12].

Nurses should map the challenges that each individual experiences so that care is individually adapted, which may lead to a change in ostomy patients’ challenges and psychological reactions over time [13].

A qualitative study from England showed that psychological problems resulting from having a stoma can occur in patients with inflammatory bowel disease; however, the patients also reported positively about living with a stoma [14]. Peers in the same situation have been observed to promote security and motivation among themselves to undergo surgery earlier. The participants felt that they had undergone a renewal of themselves after the installation of a stoma, which gave them a feeling of liberation and helped them reclaim themselves [14].

Conversely, a study from Singapore stated that stoma patients are not prepared to handle the stoma when they are in the hospital, which affects their competence in stoma care after the operation. The study also showed that several patients experienced psychological and physical challenges regarding the stoma and needed assistance from health personnel, especially stoma nurses, for improved training and guidance [15]. This is in accordance with the findings of Alenezi et al. (2022), who showed that counselling and guidance from competent nurses before and after the surgical stoma surgery are of crucial importance for better quality of life [16].

In our study, we chose to focus on women mostly due to our specific interest in how it might be experienced for younger women to live with a stoma. At the age between 20 and 35 years, most women go through changes in life, both mentally and physically, regarding fertility, pregnancy and having children. We wanted to investigate particularly in terms of fertility and sexuality how a stoma affected their lives.

To the best of our knowledge, no qualitative Norwegian study has explored younger women’s experiences after having a stoma fitted. Although studies have shown how living with a stoma can bring both challenges and positive outcomes and what measures can help, we have little information about how young women with inflammatory bowel disease and a stoma experience intimacy and everyday life. Studies with young, female ostomy patients’ experiences related to how they are met by nurses are also lacking.

The purpose of this study is to investigate how younger women experience living with an ostomy in order to examine, from a patient perspective, what healthcare personnel can do to help ensure that this patient group feels cared for. We chose the following research question: How do younger women experience intimacy, sexuality and everyday life living with a stoma?

## 2. Materials and Methods

This study has a qualitative design and uses a descriptive and explorative approach to investigate how younger women experience living with a stoma. Individual in-depth interviews were conducted to shed light on the study’s research question. Individual in-depth interviews are suitable when it comes to topics that can be perceived as sensitive and private [17]. This study employed the COREQ checklist.

### 2.1. Selection of Participants

The study’s inclusion criteria were women aged between 20 and 35 years, living in Norway and having different types of stomas, both colostomy and ileostomy. For reasons of anonymity, the educational background/occupation and the number of children the participants have were not disclosed. A secretary working at The Norwegian Association of Ostomy, Reservoir, and Intestinal Cancer (NORILCO) and a secretary working at a surgical gastroenterology department at a local hospital helped with the recruitment of participants by email inquiry. The two secretaries sent emails to former patients who met the inclusion criteria. None of the women who were contacted refused to be interviewed, and none of them dropped out during the study. Throughout the recruitment process, we found that it was difficult to obtain relevant candidates who fit the inclusion criteria, as ostomy patients constitute a small patient group. We tried to gain access to recruit from a larger regional hospital but were rejected.

### 2.2. Data Collection

Between February and June 2021, six interviews were conducted by a female, skilled master’s student in nursing (AEM) who was a practicing nurse working in homebased nursing. Before performing the qualitative study, the researcher received methodological education in a descriptive and explorative approach and training in interview techniques. Four women participated, of which two were interviewed twice. The participants who were interviewed twice were going through a change with their stoma, which was of interest to conduct further follow-ups. Due to the COVID-19 pandemic, the interviews had to be conducted via online video calls using Zoom and were digitally recorded.

The interviews were transcribed by the first author who had been given instructions in how to transcribe from her supervisor. Body language, pauses, laughter and sounds were added to the transcriptions to make the data material as close to reality as possible. Neither the transcripts nor the findings were returned to the participants for comments or correction, but the researcher often asked the informants if she had understood them correctly during the interviews. A semi-structured interview guide was developed and used during the interviews. The questions from the interview guide concerned topics such as coping with everyday life, work, social network, diet, self-image, intimacy, sexuality and follow-up by healthcare personnel. The guide was based on our own preconceptions and previous research. The starting point for the interviews was a relaxed and light atmosphere, the researcher talked about her professional background and her goal with the study, which contributed to an open conversation. During the interviews, key words and comments were noted to provide a basis for reflection during the analysis process, as well as to gain an overall impression as close to reality as possible. At the end of the interviews, a summary was prepared about the participants’ experiences with the interviews and whether they wanted to add anything beyond what we had talked about. The interviews lasted between 45 and 70 min.

### 2.3. Analysis

Throughout the analysis, both authors used a text condensation method [17] based on the individual experiences the participants conveyed through the interviews. Malteruds’ analysis method consists of four systematic and structured steps, in which both co-authors participated. We had a need to engage in a back-and forth process of our reflections, and we had to be close to the data to discuss the findings further and thus improve transparency. Additionally, given the volume of data generated from merely six interviews, we considered that the analysis could be done manually.

In Step 1 (Overall impression), the interviews were transcribed and read through several times to gain an overview of the data material. Keywords and preliminary themes were identified, e.g., challenging taboo topics, information, self-image and social relations. In Step 2, meaning units were identified, and provisional themes were looked at more closely and converted into codes (Table 1).

The data material was systematised using colour codes. In Step 3 (Condensation), the material was sorted and inserted into tables. Possible subthemes as ostomy nurses’ contribution, time for preparation and ignorance from most people were created and assessed. In Step 4 (Synthesis), quotations that should represent the various subthemes were selected together with the theme. A summary of the analysis was performed, and the findings were validated, with an assessment of whether the findings were faithful to the interviews and the transcription.

### 2.4. Ethical Considerations

The study was approved in advance by the Norwegian Center for Research Data (NSD), project no. 684713, and the Regional Committees for Medical and Health Research Ethics (REK). Dictaphone via ‘Nettskjema’ was used as a sound recorder for the interviews. The recordings were deleted after the transcription was completed. The transcription was anonymised during the process. An information letter was sent to the participants in advance, and all gave consent to participate in the study. The informants were informed about the possibility of withdrawing from the study at any time without reason.

## 3. Results

The participants had different starting points; some of them had a permanent stoma, and others had it only for a period. All participants’ underlying causes were individual. Among the participants, some also had a reservoir in addition to a stoma, some had to replace the temporary stoma and others had to later choose between a permanent stoma or a reservoir. The participants had either Crohn’s disease or ulcerative colitis. Some of the participants had experienced disease symptoms in early childhood and had thus lived with the disease for many years, whereas others were diagnosed later in their teens. All participants had been diagnosed for several years. The participants had different types of stomas, both colostomy and ileostomy. One of them had also tested out the pelvic reservoir when she was younger but switched to a stoma as an adult. The women’s ages varied from 14 to 25 years at the time they had a stoma. They lived in different parts of Norway, both in urban and in rural areas.

The findings are divided into three themes, with each theme having a corresponding subtheme or subthemes derived from the analysis: (1) The importance of follow-up and information from healthcare personnel, (a) Ostomy nurses contribute to support and security, (b) Follow-up and information depend on the place of residence and (c) Silence about challenging taboo topics such as sexuality; (2) Illness experiences and freedom in everyday life, (a) Time for preparation in advance and (b) Went better than expected and (3) Self-image and social relationships, (a) Became more confident in themselves and (b) Faced ignorance from most people.

### 3.1. The Importance of Follow-Up and Information from Healthcare Personnel

We observed that the participants mostly felt cared for in a dignified and respectful way by health personnel. However, one participant said that her dignity was not taken care of by nurses when she was hospitalised in a four-person room. She stated the following:

*“Some of the nurses thought there was such a nasty smell from my stoma, so they had to wheel me out of the room before emptying it. I can see what they mean by it being a strange smell as it smells different to normal poo, at least ileostomy. Eh, but they didn’t wheel the other people who had to poo with their bottoms, say on a chair or a basin out of the room, so I think it was a bad way to do it when you’re a healthcare personnel, that they made it a more disgusting thing than it really was.”* (P1)

The participant was treated in an undignified manner where lack of respect for privacy and shielding compared to fellow patients made her feel disgusting, which could be experienced as shame and rejection.

#### 3.1.1. Ostomy Nurses Contribute to Support and Security

The participants who received follow-ups from a stoma nurse expressed that this contributed to their feelings of security. They received instructions about stoma care and gained useful knowledge about diet and which foods one should be careful about testing out in the initial period with a stoma. The ostomy nurse also informed them of the various challenges that may arise. According to the participants, one of the greatest advantages of having a stoma nurse was that they gained knowledge about different equipment that could be adapted to each individual stoma, which helped them, for example, to avoid leakage. The participants were also grateful that the stoma nurses they were in contact with showed an understanding of their situation, gave useful advice and were available. Moreover, they were invaluable conversational partners if the participants did not want to or did not feel safe talking to family and friends. One of the women stated the following:

*“After being a patient for ten years, I am used to meeting nurses, and I experience the stoma nurse a little closer than other nurses. After all, she is sitting and cleaning my bowel, and suddenly it starts pooing while she is changing it. Fortunately, that’s her job, so it’s not embarrassing because I know she sees many other stomas.”* (P3)

The woman put into words the importance of being followed up by experienced stoma nurses who contribute to their confidence in an uncomfortable and intimate situation. The findings also indicate a contrast between nurses on bedside duty and stoma nurses in that the participants do not have the same confidence that they have the level of expertise in stoma care and in providing useful information to patients.

#### 3.1.2. Follow-Up and Information Depend on the Place of Residence

We found that the closer the patients live to larger cities and larger hospitals, the more thorough follow-up they receive from stoma nurses and other healthcare personnel. This means that the participants found information on the internet, became members of groups on Facebook or contacted others who had a part of their colon removed to gain insights into the everyday life with a stoma. One participant said that she experienced poor follow-up when she lived in a place with a smaller hospital, and after she moved to a new place, she experienced good follow-up from healthcare personnel. She stated the following:

*“After I moved, I received a completely different follow-up. Then it was a matter of course that I would come to the ostomy nurse, and we would look at equipment and get follow-up and find something that suited me. Where I lived before, and because I couldn’t find the right product, I had a lot of leaks.”* (P2)

It emerges that living in a rural area where healthcare personnel have insufficient knowledge of adequate ostomy equipment contributed to frequent leaks and potentially undignified situations with discomfort for the skin, impurity on clothing and social isolation. Some of the participants also said that they felt well taken care of by healthcare personnel at large specialised hospitals, precisely because they had extensive experience with ostomy patients and the capability to follow up with new patients.

#### 3.1.3. Silence about Challenging Taboo Topics such as Sexuality

The participants expressed that they had several questions about having a stoma, not only about how one changes and relates to the stoma itself but also about topics such as sexuality and cohabitation. Such topics can be challenging for patients to ask about, as such experiences are private and can feel embarrassing to talk about. The participants had mixed experiences with how healthcare professionals provided information on these topics. Some received no information, and there was very little room to talk about sexuality related to a stoma, whereas others revealed that healthcare personnel only talked about the topic if the patient brought it up. All participants expressed that they had questions about stoma and cohabitation. However, the information they received was very scarce, and sexuality related to the stoma was rarely thoroughly reviewed. The analysis of the interviews also showed that three participants did not receive information regarding fertility and stoma. One participant stated the following:

*“A stoma nurse once told me bluntly, “You can’t expect to get pregnant,” and that’s what I got as information after the operation. So far, it had nothing to do with the stoma but the construction of a pelvic reservoir. I was 13 years at the time, so that topic wasn’t relevant, but I don’t understand how you can talk about such a topic in such a situation.”* (P1)

This quote seems to imply that information is important, but the timing and sensitivity are even more crucial in addition to adaptation to each individual patient and each individual situation.

### 3.2. Illness Experiences and Freedom in Everyday Life

The analysis showed that advanced preparation contributes to an easier acceptance of the new situation by patients. However, we noted that the participants who did not have time to prepare nevertheless experienced that living with a stoma went better than expected.

#### 3.2.1. Time for Preparation

The participants’ experiences and handling of having a stoma were quite individual. The difference was significantly large if a stoma was fitted acutely or via planned surgery. An acute stoma gives little time for preparation, and the participants expressed that when they woke up with a stoma on their stomach, they were unsure of what would happen next and what their new everyday life would be like. Whether the stoma is fitted only for a period or is permanent also had an impact on how they handled the situation. The participant who received a temporary stoma accepted the new everyday life relatively quickly, while those who received a permanent stoma needed more time to accept the situation.

The participants had Crohn’s disease or ulcerative colitis with varying disease durations. The analysis revealed that their time before the stoma was characterised by severe or almost unbearable abdominal pain, bloody stools and weight loss, which often led to hospitalisation. The participants were tired and spent a lot of time indoors, near a toilet. Some were also bedridden because of the symptoms of the disease. Three of the participants had received strong drug treatment for a long time to alleviate their symptoms, but their bodies could no longer tolerate it. The long-term treatment led to the doctors recommending them a stoma. Some of the informants received the message at a young age and were therefore sceptical about saying ‘yes’. Others had not heard of a stoma before it was brought up by the doctors, and one of the women said the following:

*“I never got a good explanation about what it was when I was a child. So, I had the horror scenario of the intestine hanging in a carrier bag; to me, that’s what an ostomy sounded like.”* (P2)

The woman described the fear of the unknown. As she received very superficial information about ostomies as a child, she did not know what an ostomy was, which made her choose to delay the stoma operation. Other participants believed that a stoma would lead to stronger and more constant pain; therefore, they chose to wait to get a stoma until it was absolutely necessary.

#### 3.2.2. Went Better than Expected

Although the participants stated that they would prefer to avoid getting a stoma, they experienced greater freedom in everyday life after having a stoma because they were no longer dependent on a nearby toilet and the pain decreased for three of them. The women said that they would probably have agreed to get a stoma earlier if they had known what it was like to live with a stoma. One participant explained that she was grateful for the time she had to process the fact that she was going to have a stoma and that she then had good contact with health personnel and the stoma nurse, who provided thorough information. After the operation, she was aware that it was normal to feel tired and knew what the stoma would look like; accordingly, she could further focus on training back into everyday life. Another woman said the following:

*“All in all, I would say it has gone much better than expected, despite my challenges… there is so much equipment. I am completely fascinated by everything that exists. Fortunately, I have felt that I have found what suits me, instead of feeling that ‘this is what we have, and this is what to use’”* (P4)

Guidance on using the equipment prevented challenges related to the stoma, which contributed to a better experience than she had imagined. It turned out that it was the disease that affected the women’s everyday lives the most and not the stoma, and the symptoms of the disease took over on bad days and had an impact on their condition and state of health, which the following quote illustrates:

*“It is a positive change compared to the illness I had before [the stoma operation] and that my appetite has returned. In a way, I feel like I’ve got my life back. I feel that all in all it is positive, I have a little more control in everyday life.”* (P4)

Getting a stoma was like a decisive ‘before and after’ for the women, in that they had better health, became more optimistic and had a more manageable everyday life. Beyond that, the participants asserted that there were only a few limitations to having a stoma. One participant stated the following:

*“You can skydive, you can dive, you can have sex, you can do anything, really.”* (P1)

Even though she was initially sceptical about being able to have a normal life with a stoma, she described the freedom it entailed. Based on the participants’ starting points, there are more advantages to having a stoma than disadvantages.

### 3.3. Self-Image and Social Relationships

A reason why more people chose to wait to get a stoma until it was necessary was ignorance of what a stoma entails. The participants learned that only a small number of people in their own environment had knowledge about a dislocated bowel and the implications of having a stoma.

#### 3.3.1. Became more Confident in Themselves

All of the participants have had a stoma for quite a long period and have handled it differently from time to time. Those who received a stoma as a teenager were unsure of themselves and felt different from others their age, whereas those who received a stoma after the age of 20 were afraid that their self-image would be negatively affected. However, the participants stated that they became more confident in themselves after having a stoma. The participants who received a stoma in adulthood said they were grateful to be able to wait, as the delay may have had an impact on how they dealt with the situation. One woman stated the following:

*“I felt I was very lucky to be able to get the permanent stoma until I was 19 and a bit more confident in myself than when I was 13.”* (P1)

Making choices and being involved as much as possible proved to be important for the participant in how she dealt with having a stoma. Some participants said that they were worried about leakage, which at times could happen several times a day; therefore, they stayed at home most of the time. Those who struggled with leakage because of ill-fitting equipment had a different experience of having a stoma later. They then received thorough follow-up by an ostomy nurse with a wide range of equipment adapted to their stoma, which led them to feel more confident in themselves and helped them gain greater control over their stoma.

#### 3.3.2. Faced Ignorance from Most People

Ulcerative colitis and Crohn’s disease are illnesses that are not necessarily visible to others, and it can also be difficult for others to understand and manage their behaviour in such situations. For example, getting advice to go for a walk or eat a little did not help the chronic stomach pains that these women experienced, and it made them feel lonely about their illness. The analysis also showed that intimate and sensitive situations, such as intercourse, became even more vulnerable with a stoma and could affect one’s self-image. One woman stated the following:

*“I’ve had a one-night stand a couple of times, where I’ve sometimes talked about the stoma in advance, sometimes not, and never experienced anything negative. Had these types of situations where you are in a very vulnerable intimate situation, regardless of whether you have a stoma or not. If I had experienced some negative comments in such a situation, I think it would have greatly affected my self-image.”* (P1)

Regardless of being open or not about the stoma, these experiences illustrate that one can be fortunate to be met in a decent way when daring to expose oneself. Three of the participants discussed incidents in which curious children commented on the stoma; in some cases, adults also stared at the stoma and questioned what it was. In most situations, strangers were the ones who made glances or comments and rarely close friends or family. At the same time, the participants stated that it can be a pleasant experience that people are curious if they ask politely; however, being obviously stared at was experienced as unpleasant. Even though three of the participants experienced unpleasant attention related to the stoma, they became more confident in themselves afterwards. One participant experienced the following misconceptions from a friend:

*“I think people should know a little more about what it really is. I have had many encounters with people displaying pleasant curiosity, and I have a very visible scar that is right next to the bag. A friend of mine asked quite seriously, ‘Is it something from where poop comes out? From this tube?’ Most people don’t know how it works. I thought for quite a long time that the intestines are pulled out and put inside the bag because they could not stay inside the body. After all, a stoma sounds like the whole intestine has been pulled out.”* (P2)

The situation emphasises that it is not only people with a stoma who need information but also the general population to reduce stigma.

## 4. Discussion

The findings indicate that follow-up and information have an impact on how young women handle having a part of their colon removed. We noted that ostomy nurses provided support and security to the participants and that the women generally experienced being met with dignity and respect by healthcare personnel. The participants who received treatment at smaller hospitals received little or no follow-up from the stoma nurse, which meant that they had to find other ways to obtain information. Furthermore, the women missed that health personnel took the initiative to talk about intimacy, sexuality and fertility with sensitive timing. Some of the participants also pointed out that the information regarding cohabitation, sexuality and intimacy was rarely shared by healthcare personnel and that they had to bring it up themselves if they had questions about these topics.

However, nurses should also take the initiative and talk to patients about challenging topics, such as sexuality [18]. It may be embarrassing and private for some patients to bring up such topics, but if nurses include a conversation about sexuality as part of stoma treatment, it can contribute to a sense of security that enables the patient to more openly express any questions or challenges related to intimacy and sexuality. Mick, Hughes and Cohen (2004) developed the assessment tool ‘BETTER’ to help nurses talk about sexuality with their patients. The use of this model also creates the opportunity for stoma patients to discuss what sexuality means to them, which makes it easier to identify any concerns or problems related to intimacy, closeness and the stoma. Moreover, talking openly about topics that can usually be experienced as taboo and private can open the door to better communication with this patient group [19].

The participants in our study stated that having a stoma could lead to a feeling of loneliness, as they knew neither the process of having parts of their bowel removed nor anyone with whom they could exchange their experiences. This study found that a stoma provides physical confirmation that one is different, especially in the initial phase of having a stoma, and leads to negative feelings. Research shows that it is individual whether ostomy patients choose to hide or show the stoma, based on embarrassment and the possibility of being accepted or rejected by others [2]. Moreover, it appears that patients between the ages of 13 and 18 want to ‘become normal again’ in order not to be perceived as different [20]. However, it appears that the participants in our study learned to live with the stoma and accepted themselves again as time went on, and the need to hide the stoma from others was no longer present as much as before. Furthermore, the school health service should emphasise teaching about diversity, identity, boundaries, anatomy and sexual health [21]. Such lessons can contribute to children and young people being exposed to inequalities in society, which might reduce the feeling of being different. In this case, associations such as NORILCO play an important role by making ostomy visible, especially for future patients who can benefit greatly from their visibility work, including through groups of like-minded people with ostomies [22]. Ostomy nurses and general nurses have a critical role to play in making such associations visible.

One of the participants in our study stated that, as a 13-year-old, she was told by the ostomy nurse shortly after surgery that the possibility of having children naturally was no longer possible, as she had a pelvic reservoir installed. Continuous evaluation is crucial in health pedagogy, where one considers which information should be given when [23]. In some cases, it may be appropriate to give a message orally; however, in situations where detailed and highly sensitive information is given, written material should also be provided [23]. However, in the analysis of our findings, it emerged that the information after a surgical intervention was often not satisfactory before the participants received follow-up from a stoma nurse, as it seemed that the stoma nurse was the only one with sufficient competence to guide stoma care.

One of the women experienced that her dignity was violated by nurses while she was hospitalised. According to Nåden and Lohne (2020), violating patients’ dignity can lead to the experience of not being seen and heard, and nurses must treat patients with dignity so that they feel cared for and safe. Respect and integrity are also fundamental elements of the concept of dignity and are fundamental when dealing with patients who are experiencing illness and suffering [24]. Based on the participants’ experiences, nurses should generally focus more attention on dignity when information is given before and after stoma surgery, leading to better care of stoma patients in vulnerable situations. If we link dignity to the participants’ experience of not receiving sufficient information from nurses on bedside duty, we see that this violates their integrity and experience of security. However, nurses’ meetings with stoma patients immediately after an operation should not necessarily be characterised by large amounts of information about stoma changes and various complications [25]. It is even more important to show support by being a fellow human and convey safety, as the patient is probably in a crisis in view of his/her new situation [26].

Despite receiving little information from healthcare personnel, the participants’ overall experience of having a stoma fitted had gone better than expected. The participants knew little about the stoma before they experienced getting one themselves, and some feared severe pain, which led to them waiting to get a stoma; in addition, some participants had little time to prepare for surgery. The waiting time, on the other hand, provided an opportunity for reflection and mental preparation for some participants, where they experienced that they were able to adjust to the changes that having a stoma entails. The participants stated that they got their lives back when they got a stoma, as they were no longer dependent on a nearby toilet, and they dared to take part in social activities that they previously could not participate in, as frequent toilet visits and severe pain characterised large parts of their everyday life. For our informants, the stoma was rarely an obstacle to leading everyday life. However, it might take time to get used to the bag on the stomach. Most people live normally with a stoma and continue with previous leisure activities and work as before [27].

### Limitations

This study included four participants and conducted a total of six interviews, which may have provided limited data, but the purpose of our exploration was a deeper understanding of the phenomena and is likely transferable to other settings and situations. Moreover, the essence of qualitative research is that it has few but in-depth data [17], and saturation was achieved related to the purpose of the study, principally through several discussions and meetings during the various analytical steps. The first author has experience working in a gastro-surgery department as well as working with ostomy users in home care nursing. This might have influenced the content of the interview guide. However, the interview guide was primarily based on previous research and was used to structure the interviews, and the researcher was open to pursuing other topics that the participants brought up during the interviews. Additional research is needed to verify the findings of this study and to further test some of the hypotheses derived from our findings in, e.g., a multicentre study with validated questionnaires.

## 5. Conclusions

In light of the results obtained, living with a stoma is a highly individualised experience that is based on age, starting point, experiences with other people and follow-up by healthcare personnel. An ostomy can present challenges, such as leakage and pain, and provides a visible confirmation of being different, which for some can be difficult to accept. At the same time, we suggest that having parts of a bowel removed can be experienced as relief, especially when the disease has previously contributed to poor self-image and social isolation. As far as we can see, there are no current national guidelines for treatment and follow-up of ostomy patients in Norway. Clearer national guidelines for the treatment and follow-up of this patient group may mean that more ostomy patients with inflammatory bowel diseases would choose a stoma earlier and thus avoid unnecessary pain because of the course of the disease. Individual and adapted information and guidance would ensure that patients understand the content and are aware of the challenges and complications that can arise with an ostomy. Making the disease visible to outsiders via information can, as we see it, lead to less stigmatisation. Greater acceptance in society may have an impact on the sexuality and cohabitation of people with a stoma, which may make it less intimidating to go on a date as the focus is shifted away from the fear of rejection due to the stoma.

## Figures and Tables

**Table 1 ijerph-20-05627-t001:** Excerpt from the analysis process.

Meaning Units	Codes	Subthemes	Themes
‘I had several conversations with the ostomy nurse prior to the operation, which meant that I had more to adapt to before the operation. It allowed me to focus more on training afterwards and not on getting to know how a stoma works, and I think that was very positive.’	Time for adaptation Could focus on training	Ostomy nurses contribute to support and security	The importance of follow-up and information from healthcare personnel
‘I hadn’t received any follow-up from the ostomy nurse, or, after all, she came by once or twice, but I lived quite deserted, so I had no one to deal with. I just had to order the equipment which the ostomy nurse said I should have, which didn’t work at all.’	Little follow-up Residence No one to deal with	Follow-up and information depend on the place of residence	
‘As in my case, I would like to have been explained ‘what does it mean to draw a stoma’ or just explain or just see what a bag looked like, feel a little what it was like to wear on the skin. I understand that it can be difficult to carry out in emergency cases, but for people who have a planned stoma, I think preparation means everything.’	Preparations, mentally and physically	Time for preparation in advance	Illness experiences and freedom in everyday life
‘There were also a lot of hospitalisations, so my everyday life was completely different when I got a stoma. I didn’t recover from having a stoma, but at least everyday life became much more predictable…’	More predictable everyday life	Went better than expected	
‘…also I may have become a bit more confident in my own body, yes, or I have gotten to know my body better.’	Got better acquainted with her own body	Became more confident in herself	Self-image and social relationships
‘I’ve always had a lot of nice bathing suits compared to other people who buy bikinis, so when I was going to the swimming pool after I had a stoma, I remember thinking “oh, why do I only have bathing suits? Now it would have been a bit of fun to wear a bikini.”’	Would like to provoke by showing off her stoma	Faced ignorance from most people	

## Data Availability

Data generated during the present study cannot be shared due to the need to preserve the informants’ privacy and confidentiality.

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
