# Peer review of "A Qualitative Study on How Younger Women Experience Living with an Ostomy"

_ijerph, 2023, doi:10.3390/ijerph20095627_

Round 1

Reviewer 1 Report

Review.   

A Qualitative Study on How Younger Women Experience,  Living with an Ostomy

Thank you for giving me the opportunity to review this manuscript

  • A brief summary 

The aim of this research was to explore how younger women experience everyday life after an ostomy and to map what healthcare personnel can do to ensure that the patient group can feel safe and looked after. The findings resulted in three main themes: The importance of follow-up and information from healthcare personnel. Secondly, experience with illness and freedom in everyday life and finally, self-image and social relationships. The women need time to prepare before surgery, and learning to live with the stoma provides a good basis for handling the new everyday life with a stoma.

Is the manuscript clear, relevant to the field and presented in a well-structured manner? 

Yes, the manuscript is clear if the structure and layout are amended.

  • Are the cited references mostly recent publications (within the last 5 years) and relevant? Does it include an excessive number of self-citations?

There are 26 references, and they are mostly up to date; about 6 are old references, and they are more than 10 years old. There are 12 references in Norwegian.

  • Is the manuscript scientifically sound and is the experimental design appropriate to test the hypothesis?

The manuscript is partly scientifically sound, and the research approach is appropriate to the research question if modified. The eight topics in the interview guide and the result does not answer the research question (in a phenomenological way) How do younger women experience living with a stoma?

  • Are the manuscript’s results reproducible based on the details given in the methods section?

The method section is clear, but some more information is needed. The structure- sampling and participant selection should be placed before the data collection—in chronological order.

How were the participants selected? By whom? I also think that there is some language confusion in the text--…and having had their bowels removed.  I do not believe that all intestines were removed. Correct the text and terminology used. I also find it difficult to understand …it was difficult to obtain relevant candidates who fit the inclusion. There are only 4 women participating, could this be due to recruitment at a surgical gastroenterology department at a local hospital?? NORILCO has a youth section for people aged 15 to 35 years of age. The interview guide was constituted of eight topics, it seems very close to a semi-structured interview. Were the interviews audio or digitally recorded? Or were there only written notes? Were the interviews transcribed verbatim, or were they corrected? If transcribed, who made the transcriptions?  Were instructions given on how to transcribe? This information should be placed in connection to the data collection- not as now in the end of the analysis.

      The analysis is a text condensation method, according to Malterud, is descriptive and explorative method. Nothing could be connected to phenomenology. This needs to be clarified.

The results are three themes with sub-themes; 1) The importance of follow-up and information from healthcare personnel, a) Ostomy nurses contribute to support and security; 2) Illness experiences and freedom in everyday life, a) Time for preparation in advance and b) Went better than expected and 3) Self-image and social relationships, a) Became more confident in themselves and b) Faced ignorance from most people. The result summarizes the words from the interviewee, and then there are quotations added repeating. I think that the analysis needs to be reworked. For instance, Faced ignorance from most people and, then there are positive attitudes and positive experiences quoted,

Four (4) women were interviewed but there are quotations from only three, and one of them has the majority of quotations.

·        Are the figures/tables/images/schemes appropriate? Is the data interpreted appropriately and consistently throughout the manuscript?

The figures/tables are clear.

Are the conclusions consistent with the evidence and arguments presented?

      In the discussion, the authors highlight sensitive issues, sexuality etc—yes that is mentioned in some quotations but not taken into account in the analysis.

      As I understand NORILCO is working on these issues as well, love, friendship, work etc.

I am pleased that there are limitations presented. But there need clarifications, saturation- how was that reached?  Clarification of the interview guide. Is nursing practice enough for qualitative analysis? Please inform the readers.

Conclusions, it feels as if the authors are jumping to conclusions. Yes, it is interesting, but I cannot see all this in the result.

  • Please evaluate the ethics statements and data availability statements to ensure they are adequate.

The ethics statements are clear.

This could be an interesting and important paper, there is a need for clarification and analysis and analysis according to the method.

The paper needs to be reworked and re-written, presenting a solid analysis which will give a stable finding to present. In the discussion, there should be a discussion about their own results compared with others and literature references, and this could be optimal if the findings/result presented these issues discussed.

I look forward to reviewing this manuscript again.

My best wishes

Reviewer 2 Report

Summary:

Manuscript ijerph-2237189 is a qualitative research study evaluating the quality of life and experience of young women having undergone intestinal ostomy placement. The study describes the results of structured interviews conducted with 4 women who underwent ostomy placement for inflammatory bowel disease and their experience with home ostomy nurses support and overall adaptation to this new condition. The authors conclude that healthcare workers should provide individualized support to patients with stomas, and that their help seems to be associated with positive experiences overall. 

Criticism:

Study design: 
The study described qualitative data obtained from 4 patients. While the study design is good, the number of patients included in this study seems very small to draw any generalizable conclusions. Some analysis should also include the age at which their disease was diagnosed, what type of stoma they have (ileostomy, colostomy, loop ileostomy, end colostomy, mucous fistula, etc.), as it may affect psycho-social, care and medical problems (amount of stool passage, risk of dehydration, etc.). The rational was including only young women and not both genders should be further described and explained, as both genders should be analyzed equally and potential body image, impact of sexuality and social life evaluated for both. 

Introduction and discussion:
The introduction is very thoroughly described and could be shortened. The discussion addresses many different aspects of care, body image, psychosocial aspects of having a stoma in a thorough and critical fashion. Limitations should be further described, especially from a sample size / cohort size standpoint, and how the authors decided to interview 4 and not 10 or more patients? 

Patients and methods: 
This section is well described, though some of the details should be moved to results, especially patients details and description of the cohort. 

Results: 
The results are well described and could be further synthesized and shortened. Some of the important questions/answers could be summarized in a table. 

Conclusion: 
The study ijerph-2237189 describes the results of a qualitative study of 4 young women who live with an ostomy and the impact this condition has on their quality of life, psychosocial interactions and overall healthcare support and experience they had with the preparation for life with a stoma and daily life with it once placed. Unfortunately, though a rare condition, the number of patients in their cohort seems very small, too small to reach any significant generalizable conclusions.

Reviewer 3 Report

First of all, I want to congratulate the authors for their work.

Here are my comments:

The study has limitations. The authors managed with the quality design to bring an insight of the quality of life of the younger women living with an ostomy.

I want to ask the authors if they consider that they work has no influence from the Covid-19 preventions measures(egg isolation) on the interviews. Mainly the patients described the past but the presence in the collectivity increased the number of negative experiences of not having control of the stoma content.

Reviewer 4 Report

The authors in their article attempt to asses young women experience of living with the stoma.

The topic, however interesting from a clinical point of view, is merely touched in this study. Moreover, the manuscript can be improved in multiple aspects.

Abstract: to make it more transparent and accessible I encourage dividing the abstract into paragraphs corresponding with the main parts of the article.

Introduction: the authors point out data from Norwegian sources, while there is a great deal of most credible data from around the world and there are numerous studies about ostomy patient's QoL. Authors should pay more attention to introduce varied evidence in the introduction in order to provide better overwiev of the topic. There are few high quality references introduced in the manuscript. 
Authors state 'no qualitative Norwegian study has explored younger women’s experiences after having a stoma fitted' which in my opinion is not a sufficient justification for the study; 1st - no qualitative study is a modest reason since these kind of study rarely provide extensive data and scientifically sound evidence; 2nd - the question should be, whether there are any studies concerning this topic.
Furthermore lines 91-94: 'The purpose of this study is to investigate how younger women experience living with a distended bowel in order to examine, from a patient perspective, what healthcare personnel can do to help ensure that this patient group feels cared for.' I can't really understand what authors had in mind by 'distended bowel' - does this stand for patients with an ostomy?

Materials and Methods: the authors introduced COREQ checklist wich is a proper tool for this kind of study. The person conducting the interviews also has good qualifications. 
Authors selected women 20-35 y.o. with ostomy with help of NORILCO and a local Surgery Department. It is unclear wheather these women are patients of the local hospital or are these all suitable candidates for the study in the entire Norway.
A group of 4 patients is very small and should be increased i.e. by conducting a multicenter study.
Qualitative studies with interviews and phenomenological approach are less standardised therefore provide less credible results. A validated questionnaires concerning qol could be used i.e. health-related quality of life (HRQoL) and acceptance of illness scale (AIS).
In order to increase the value of the study a comparison between a group young women and general population of ostomy patients should be made. 

Results: overall contain some interesting key points which could be a possible roadmap for ostomy caregivers, however the quality of evidence is very low due to the methods used and the number of studied subjects. 

Discussion and conclusions: while forming conclusions authors use strong words, like 'the findings show', 'study highlighted', 'we showed that' - I advise being more careful in giving conclusions in this form since little can be shown based on 4 people experience - another 4 people could have had very different opinion.
Starting line 460 authors claim: 'Clearer national guidelines for the treatment and follow-up of ostomy patients may mean that more ostomy patients with bowel diseases, such as ulcerative colitis or Crohn’s disease, would choose a stoma earlier and thus avoid unnecessary pain because of the course of the disease.' What is this sentence based on? What are the current guidelines on the treatment and follow-up of ostomy patients that are unclear? Authors should specify their suggestion. 

References: the reference format is non-uniform and should be revised - add DOI if applicable - see the Journal guidelines. I encourage adding more references of high value - clinical trials, meta-analysis, systematic reviews - for providing background of the study and to support your statements.

OVERALL

The authors correctly identified the limitations of their study, however in my opinion the study sholud be expanded in order to be suitable for publication as an original article - increase the number of participants, gather a control group, introduce standardised tools. 

Round 2

Reviewer 1 Report

Thank you for all your efforts in amending this manuscript.  Most comments from the reviewers have been taken into consideration.

Just a triviality to notice; you state that the inclusion criterion was, had removed a part of the colon, but in the result, you state The participants had different types of stomas, both colostomy and ileostomy. Please correct

Author Response

Thank you for pointing out this important detail! We have corrected accordingly (page 4, line 118).

Reviewer 4 Report

Thank you for your answers. I hope you found my comments valuabe.

I wholeheartedly advise continuing the study on the topic with a greater number of participants and with the use of unified tools. It is difficult for me to comprehend the lack of cooperation from another hospital you mentioned. I encourage you to consider expanding the study to other countries and perhaps to form an international team.

Author Response

Your comments were valuable, thanks again!

Thank you for your advice and your engagement! Yes, it is a good idea to expand our study to other countries, and we are discussing forming a Scandinavian team.